# ATR is a multifunctional regulator of male mouse meiosis

Alexander Widger[1], Shantha K. Mahadevaiah[1], Julian Lange[2], Elias ElInati[1], Jasmin Zohren[1], Takayuki Hirota [1], Sarai Pacheco [3,4], Andros Maldonado-Linares[3,4], Marcello Stanzione[5], Obah Ojarikre[1], Valdone Maciulyte[1], Dirk G. de Rooij[6], Attila Tóth[5], Ignasi Roig [3,4], Scott Keeney [2] & James M.A. Turner[1]

Meiotic cells undergo genetic exchange between homologs through programmed DNA double-strand break (DSB) formation, recombination and synapsis. In mice, the DNA damage-regulated phosphatidylinositol-3-kinase-like kinase (PIKK) ATM regulates all of these processes. However, the meiotic functions of the PIKK ATR have remained elusive, because germline-specific depletion of this kinase is challenging. Here we uncover roles for ATR in male mouse prophase I progression. ATR deletion causes chromosome axis fragmentation and germ cell elimination at mid pachynema. This elimination cannot be rescued by deletion of ATM and the third DNA damage-regulated PIKK, PRKDC, consistent with the existence of a PIKK-independent surveillance mechanism in the mammalian germline. ATR is required for synapsis, in a manner genetically dissociable from DSB formation. ATR also regulates loading of recombinases RAD51 and DMC1 to DSBs and recombination focus dynamics on synapsed and asynapsed chromosomes. Our studies reveal ATR as a critical regulator of mouse meiosis.

[1] Sex Chromosome Biology Lab, The Francis Crick Institute, 1 Midland Road, London NW1 1AT, UK. [2] Molecular Biology Program, Howard Hughes Medical Institute, Memorial Sloan Kettering Cancer Center, New York, NY 10065, USA. [3] Genome Integrity and Instability Group, Institut de Biotecnologia i Biomedicina, Universitat Autònoma de Barcelona, Cerdanyola del Vallès, Barcelona 08193, Spain. [4] Department of Cell Biology, Physiology and Immunology, Cytology and Histology Unit, Universitat Autònoma de Barcelona, Cerdanyola del Vallès, Barcelona 08193, Spain. [5] Institute of Physiological Chemistry, Faculty of Medicine at the TU Dresden, Fetscherstraße 74, 01307 Dresden, Germany. [6] Center for Reproductive Medicine, Academic Medical Center, University of Amsterdam, Amsterdam 1105 AZ, The Netherlands. Correspondence and requests for materials should be addressed to J.M.A.T. (email: james.turner@crick.ac.uk)

A TR is a serine–threonine kinase with ubiquitous functions in somatic genome stability and checkpoint control[1]. Studies on non-mammalian organisms have revealed that ATR is also essential for meiosis. ATR orthologs regulate meiotic double-strand break (DSB) resection[2], stoichiometry of DSB-associated strand-exchange proteins RAD51 and DMC1[3],

inter-homolog bias[4, 5] and crossover formation[6]. They are also components of prophase I checkpoints that ensure centromere pairing[7], timely repair of recombination intermediates[8, 9] and correct coupling of DNA replication with DSB induction[10, 11]. In humans, hypomorphic *Atr* mutations cause Seckel syndrome, a pleiotropic, autosomal recessive disorder associated with

dwarfism, craniofacial abnormalities, intellectual disability and cryptorchidism[12]. In human cancer cell lines, ATR haploinsufficiency impairs the DNA damage response[13]. Determining the functions of ATR in mouse meiosis has been challenging. Heterozygous *Atr* deletion compromises postnatal survival[14] and homozygous deletion causes embryonic lethality[14, 15]. An inducible *Cre-ERT2* approach recently revealed that ATR regulates meiotic sex chromosome inactivation (MSCI), the silencing of the X and Y chromosomes in male meiosis, via serine-139 H2AX phosphorylation (γH2AX)[16]. However, this method resulted in partial rather than complete ATR depletion.

Here we describe a superior conditional strategy for dissecting additional meiotic ATR functions. Using this approach, we show that ATR regulates homologous synapsis as well as multiple steps in recombination. By generating mutants deficient in both ATR and ATM, we identify shared and distinct functions for these kinases in mouse meiosis.

## Results

**A strategy for efficient meiotic *Atr* depletion**. For this purpose, we generated male mice carrying one *Atr* floxed (*Atr^fl*) allele, in which the exon 44 kinase domain of *Atr* is flanked by *loxP* sites[17], and one *Atr*-null (*Atr^−*) allele, in which the first three coding exons of *Atr* are replaced by a neomycin selection cassette[14]. The resulting *Atr^fl/−* males also carried a transgene expressing *Cre* recombinase under the control of either a *Stra8* or *Ngn3* promoter fragment. *Stra8-Cre* is expressed from P3 (postnatal day 3)[18], while *Ngn3-Cre* is expressed from P7[19, 20]. Testis weights at P30 were reduced three- to fourfold in *Atr^fl/−* *Stra8-Cre* males and *Atr^fl/−* *Ngn3-Cre* males relative to *Atr^fl/+* *Cre*-carrying (i.e., *Atr* heterozygous) controls, while body weights were unaffected (Fig. 1a). We observed no difference in testis weights between *Atr^fl/+* males carrying *Cre* transgenes and those not carrying *Cre* transgenes (Fig. 1 legend). Western blotting showed that ATR protein was reduced in *Atr^fl/−* *Stra8-Cre* testes, and even more so in *Atr^fl/−* *Ngn3-Cre* testes (Fig. 1b). This finding supports previous evidence that the majority of testis ATR expression occurs in spermatocytes[16, 21]. Testis histology revealed germ cell failure at seminiferous tubule stage IV, corresponding to mid pachynema of meiosis, in both *Cre* models (Fig. 1c), reminiscent of findings in *Atr^fl/−* *Cre-ERT2* mice[16]. However, the stage IV elimination was clearly less robust in *Atr^fl/−* *Stra8-Cre* than *Atr^fl/−* *Ngn3-Cre* males, because elongating spermatids were observed in some testis sections from the former but not latter genotype (Fig. 1c inset). We therefore focused on *Atr^fl/−* *Ngn3-Cre* mice (hereafter *Atr^fl/−*), with *Atr^fl/+* *Ngn3-Cre* (hereafter *Atr^fl/+*) serving as controls.

Combined immunofluorescence for ATR and the axial element protein SYCP3[22] confirmed that the characteristic ATR staining pattern observed in control leptotene, zygotene and pachytene spermatocytes (Fig. 1d) was absent in *Atr^fl/−* males (Fig. 1e). Furthermore, MSCI, assayed at early pachynema by acquisition of γH2AX on the XY bivalent and RNA fluorescent in situ hybridization (FISH) to detect absence of expression of the X-chromosome gene *Scml2*, was present in control males (Fig. 1f) but abolished in *Atr^fl/−* males (Fig. 1g). Thus, by multiple criteria, *Atr^fl/−* males exhibited efficient ATR depletion.

At stage IV, when wild-type spermatocytes reach mid pachynema, *Atr^fl/−* spermatocytes contained highly fragmented chromosome axes and nucleus-wide γH2AX staining (Supplementary Fig. 1a; see Methods for meiotic staging criteria used throughout this study). These mid-pachytene *Atr^fl/−* cells were readily distinguishable from *Atr^fl/−* cells at leptonema, in which axial elements were shorter and uniform in length, and γH2AX staining across the nucleus was more heterogeneous (Supplementary Fig. 1b). Mid-pachytene axis fragmentation and nucleus-wide γH2AX staining were also noted in *Atm^−/−* males (Supplementary Fig. 1c), as described previously[23]. Neither phenotype was observed in *Spo11^−/−*, *Dmc1^−/−* and *Msh5^−/−* males (Supplementary Fig. 1d-f), which display stage IV arrest. Instead, γH2AX in *Spo11^−/−* spermatocytes was restricted to the transcriptionally inactive pseudosex body (Supplementary Fig. 1d), while in *Dmc1^−/−* and *Msh5^−/−* spermatocytes it formed axis-associated clouds (Supplementary Fig. 1e,f), consistent with published reports[23–25]. These findings suggested that mid-pachytene axis degeneration and nucleus-wide γH2AX staining are features of ATR and ATM (ataxia telangiectasia mutated) deletion, and not merely a consequence of stage IV germ cell death.

**Mid-pachytene elimination in males deficient in *Atr*, *Atm* and *Prkdc*.** We investigated mechanisms driving mid-pachytene elimination of *Atr^fl/−* spermatocytes. In male mice stage IV arrest in response to recombination defects is promoted by ATM[26], whereas MSCI failure can cause stage IV elimination independently of ATM[16, 27]. Since *Atr^fl/−* males exhibit defective MSCI, we predicted that mid-pachytene germ cell loss would be preserved in mice doubly deficient for ATR and ATM, irrespective of whether ATR is involved in recombination.

To test this prediction, we examined testis histology in *Atr^fl/−* *Atm^−/−* mutants. As expected, double-mutant testes exhibited greatly reduced ATR and ATM protein levels (Fig. 2a, b). At leptonema, axis morphology was grossly unaffected (Fig. 2c, left panel; Supplementary Fig. 1g), but most spermatocytes at later stages exhibited axial fragmentation (Fig. 2c, right panel inset; Supplementary Fig. 1g). Thus, axis morphology was more severely compromised in double mutants than in either single mutant. Leptotene and zygotene H2AX phosphorylation are catalyzed by ATM and ATR, respectively[23, 24, 28, 29]. In leptotene

**Fig. 1** A conditional strategy for efficient depletion of ATR during male mouse meiosis. P30 testis and body weights (**a**), testis western blots (**b**), and periodic acid–Schiff and hemotoxylin/eosin-stained stage IV testis sections (**c**) in *Atr^fl/+* *Stra8-Cre* males (*n* = 9 males), *Atr^fl/−* *Stra8-Cre* males (*n* = 10 males), *Atr^fl/+* *Ngn3-Cre* males (*n* = 13 males) and *Atr^fl/−* *Ngn3-Cre* males (*n* = 13 males; means and *p* values for **a** indicated; unpaired *t*-test). Testis weights in *Atr^fl/+* *Stra8-Cre* and *Atr^fl/+* *Ngn3-Cre* males are not significantly different from those in *Atr^fl/+* *Cre* negative males derived from the same crosses (*n* = 17 *Atr^fl/+* *Cre* negative males from the *Stra8-Cre* cross, *p* = 0.37, *n* = 11 *Atr^fl/+* *Cre* negative males from the *Ngn3-Cre* cross, *p* = 0.06). Inset in **c** shows presence of elongating spermatids in some tubules from *Atr^fl/−* *Stra8-Cre* males. **d**, **e** ATR (magenta) and SYCP3 (green) staining in *Atr^fl/+* *Ngn3-Cre* (denoted *Atr^fl/+*) and *Atr^fl/−* *Ngn3-Cre* (denoted *Atr^fl/−*) males. In *Atr^fl/+* males (*n* = 2 males) ATR is observed as foci (see insets) in 85% of leptotene cells (*n* = 20 cells) and 82% of zygotene cells (*n* = 28 cells), and on the asynapsed region of the XY bivalent in 100% of pachytene cells (*n* = 30 cells). In *Atr^fl/−* males (*n* = 2 males) ATR is observed in no cells at these three stages (*n* = 21, 30 and 32 cells at leptonema, zygonema and pachynema, respectively). **f** Validation of ATR depletion in *Atr^fl/−* by analysis of MSCI. In *Atr^fl/+* males, all pachytene cells show coating of the X and Y chromosome (arrowheads) but not the pseuodautosomal regions (PAR, arrowhead) by γH2AX (magenta; left panels; XY bivalent in box magnified in smaller panels). This coating causes silencing of the X-chromosome gene *Scml2* (magenta; right panels) and compartmentalization of the XY bivalent (labeled with HORMAD2; green) in the sex body (*n* = one male, 29 cells; magnified in smaller panels). **g** In *Atr^fl/−* males, XY chromosome γH2AX coating and sex body compartmentalization do not occur. As a result, *Scml2* expression (arrow) persists in all early pachytene cells (*n* = one male; 30 cells). Scale bar in (**c**) 20 μm, other scale bars 10 μm

spermatocytes from $Atr^{fl/-}$ $Atm^{-/-}$ mutants, γH2AX was absent (Fig. 2c, left panel). Furthermore, zygotene spermatocytes that lacked axial fragmentation, and could thus be unambiguously staged, lacked γH2AX (Fig. 2c, middle panel). Thus, both phosphatidylinositol-3-kinase-like kinases (PIKKs) had been efficiently depleted in these double mutants. Consistent with our prediction, stage IV elimination was preserved in $Atr^{fl/-}$ $Atm^{-/-}$ testes (Fig. 2d). In these males we observed a population of spermatocytes with nucleus-wide γH2AX staining. Based on their advanced axial fragmentation pattern (Fig. 2c, right panel) and adlumenal location within seminiferous tubule sections (Supplementary Fig. 1h), these spermatocytes were inferred to be at mid pachynema.

Our findings were consistent with a checkpoint-independent mechanism, most likely defective MSCI, driving germ cell loss in $Atr^{fl/-}$ $Atm^{-/-}$ males. However, the remaining DNA damage-regulated PIKK, PRKDC, was still present in these mutants and could contribute a checkpoint function. We therefore examined germ cell progression in males deficient in all three DNA damage-regulated PIKKs. For this experiment we used the *Prkdc scid* mutation[30] and an *Atm flox* allele[31], because combined homozygosity for *Prkdc scid* and the *Atm*-null mutation causes embryonic lethality[32]. In $Atr^{fl/-}$ $Atm^{fl/-}$ $Prkdc^{scid/scid}$ males, testis ATM and ATR levels were depleted (Fig. 2a, b), and consequently ATM- and ATR-dependent γH2AX staining was absent (Fig. 2e, left and middle panels). Interestingly, pachytene nucleus-wide γH2AX staining was also abolished (Fig. 2e, right panel). Thus,

PRKDC mediates pachytene serine-139 H2AX phosphorylation in $Atr^{fl/-}$ $Atm^{fl/-}$ males. Nevertheless, in the triple-mutant stage IV germ cell loss still occurred, and axis fragmentation was even more severe than in $Atr^{fl/-}$ $Atm^{fl/-}$ males (Fig. 2e, right panel; Fig. 2f). Thus, mid-pachytene elimination persists in mice deficient in all three DNA damage-regulated PIKKs.

**_Atr_ regulates homologous synapsis.** We also investigated synapsis and recombination in $Atr^{fl/-}$ males, and where possible we compared findings in this mutant to those in $Atr^{fl/-}$ $Atm^{fl/-}$ males. To address whether ATR regulates homologous synapsis, we immunostained for SYCP3 and the asynapsis marker HOR-MAD2. We focused on $Atr^{fl/-}$ cells at early pachynema, i.e., prior to extensive axial element fragmentation (Supplementary Fig. 1a). Synapsis was normal in 86% ($n = 104$) of early pachytene cells from control males, a frequency similar to that in $Atr^{fl/+}$ males without the *Ngn3-Cre* transgene (89%; $n = 100$). As expected, in control cells HORMAD2 was absent on the autosomal bivalents and present on the non-homologous, asynapsed regions of the X and Y chromosome in these instances (Fig. 3a). Furthermore, in these cells, non-homologous synapsis between the sex chromosomes and the autosomes was not observed. However, in $Atr^{fl/-}$ males, only 20% ($n = 107$) of early pachytene cells achieved complete homologous synapsis. The remaining cells exhibited varying degrees of asynapsis affecting the XY pair and the autosomes (Fig. 3b; see below for details). In addition, X and Y

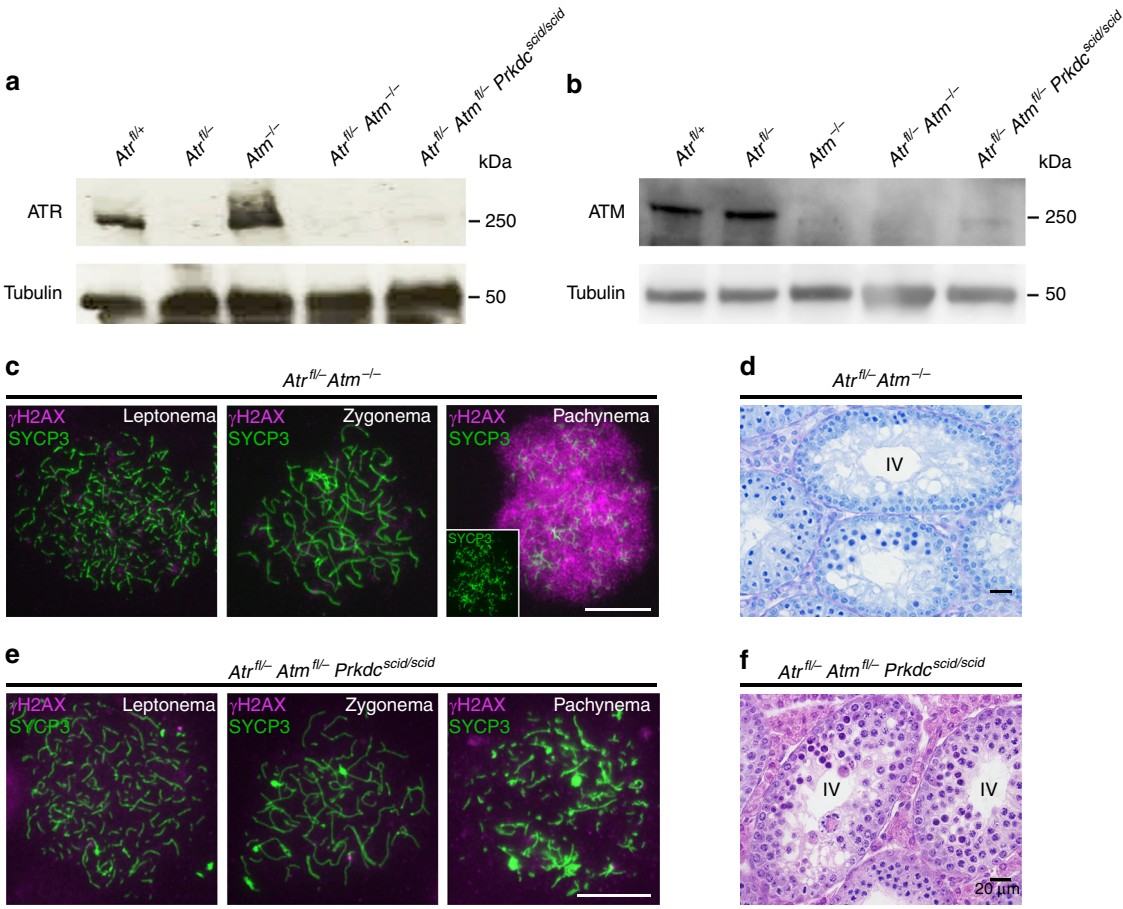

**Fig. 2** Mid-pachytene germ cell elimination is preserved in mice deficient in the PIKKs. Western blot showing **a** ATR and **b** ATM depletion in mice with different PIKK mutations. **c** γH2AX (magenta) and SYCP3 (green) immunostaining ($n = 2$ males, 25 cells for each stage) and **d** stage IV elimination in $Atr^{fl/-}$ $Atm^{-/-}$ males. **e** γH2AX and SYCP3 immunostaining ($n = 2$ males, 25 cells for each stage) and **f** stage IV elimination in $Atr^{fl/-}$ $Atm^{fl/-}$ $Prkdc^{scid/scid}$ males. Scale bar 20 μm in **d**, **f** and 10 μm in **c**, **e**

self-synapsis and non-homologous synapsis between the sex chromosomes and autosomes were more common in $Atr^{fl/-}$ cells (Fig. 3b). In mice, the XY pseudoautosomal regions (PARs) undergo late synapsis and DSB formation[33], and in yeast the homolog bias of late-forming DSBs is partially dependent on

ATR[2]. We therefore determined whether asynapsis in $Atr^{fl/-}$ males more often affects the sex chromosomes than the autosomes. We identified the sex chromosomes using DNA FISH (*Slx* probe for X and *Sly* probe for Y; Fig. 3a, b; insets). In $Atr^{fl/-}$ males, 77% of early pachytene cells exhibited XY asynapsis, while 59%

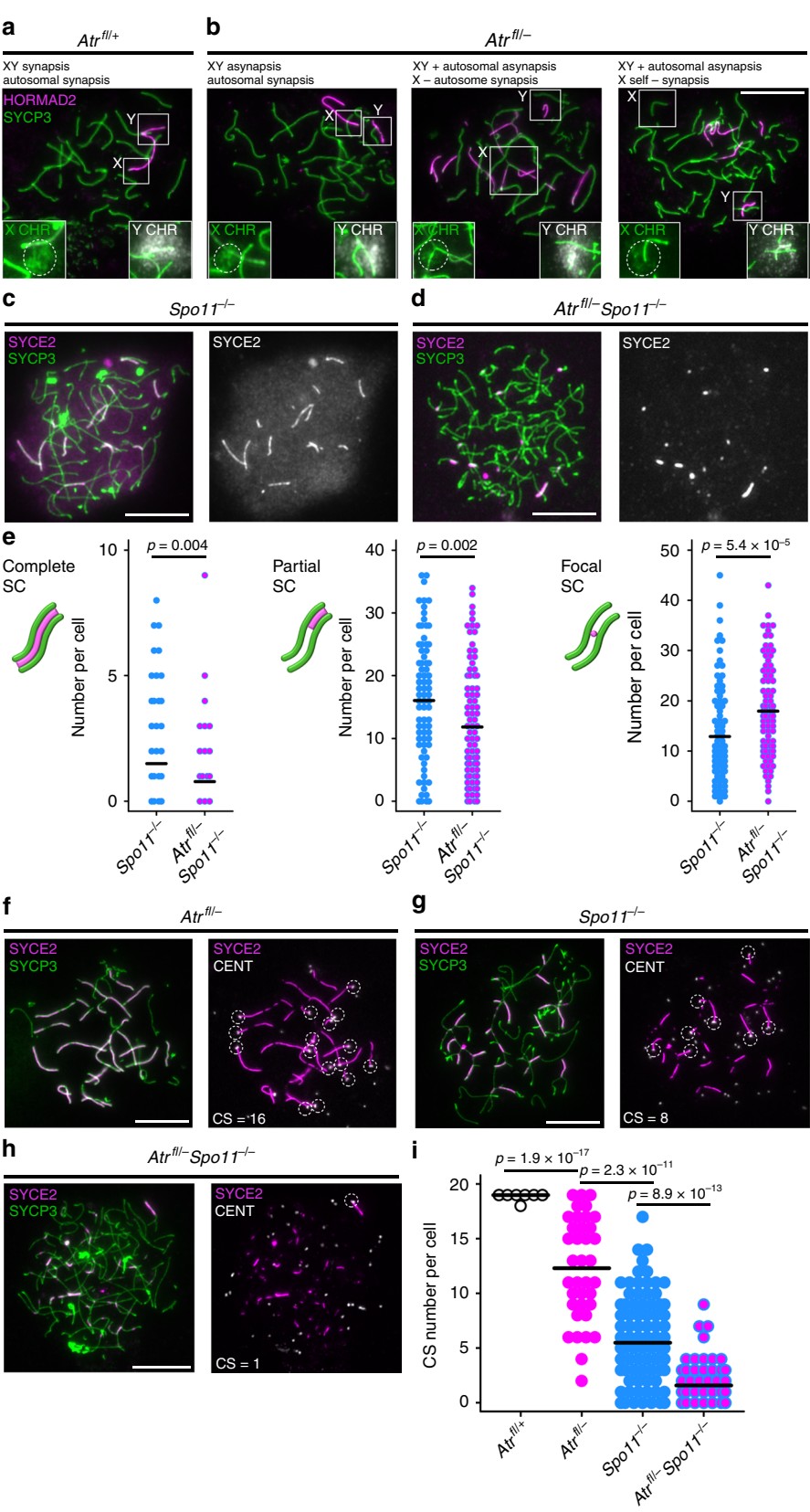

exhibited autosomal asynapsis (see legend for further details). In control males, 13% of early pachytene cells exhibited XY asynapsis and 13% autosomal asynapsis. Thus, ATR deletion has a more deleterious effect on XY than on autosomal synapsis.

Asynapsis can result from defects in synaptonemal complex (SC) formation or recombination. To address whether ATR can promote SC formation independent of recombination, we used the *Spo11*-null mutation, which permits genetic dissociation of synapsis from recombination initiation[34]. *Spo11*$^{-/-}$ males do not form programmed DSBs, yet achieve extensive SC formation between non-homologs[35, 36]. If deletion of ATR impeded SC formation in *Spo11* nulls, then ATR must have a role in SC formation. We therefore compared synapsis between *Spo11*$^{-/-}$ and *Atr*$^{fl/-}$ *Spo11*$^{-/-}$ males. Like each single mutant, *Atr*$^{fl/-}$ *Spo11*$^{-/-}$ males exhibited stage IV germ cell elimination (Supplementary Fig. 2a,b). We classified SC formation, assessed using SYCP3 and the SC central element component SYCE2[37], into three classes: (i) complete SC, encompassing the entire axis length, (ii) partial SC, extending along only part of an axis, and (iii) focal SC (Fig. 3c–e). Relative to *Spo11*$^{-/-}$ males (Fig. 3c), *Atr*$^{fl/-}$ *Spo11*$^{-/-}$ males (Fig. 3d) exhibited a decrease in complete and partial SC formation, and an increase in focal SC (Fig. 3c–e). These findings suggest that ATR promotes conversion of SC foci into longer SC stretches. Using *Atr*$^{fl/-}$ *Spo11*$^{-/-}$ males, we also demonstrated that formation of the pseudosex body in *Spo11*$^{-/-}$ males is ATR dependent (Supplementary Fig. 2c, d).

Our data suggested that ATR can promote synapsis independent of recombination. To strengthen these findings, we devised an additional method to quantify synapsis. We triple immunostained cells for SYCP3, SYCE2 and centromeres, and counted the number of centromeres that had achieved synapsis, i.e., that colocalized with SYCE2 signals (centromere-SYCE2, or CS number; Fig. 3f–i). The mean CS number was reduced in *Atr*$^{fl/-}$ relative to control males, confirming that ATR is required for normal levels of synapsis (Fig. 3f, i). In *Atr*$^{fl/-}$ *Spo11*$^{-/-}$ males, the mean CS number was reduced relative to that in *Spo11*$^{-/-}$ males (Fig. 3g–i). Thus, ATR promotes SC formation both in the presence and absence of SPO11-generated DSBs. We did not compare synapsis in *Atr*$^{fl/-}$ males with that in *Atr*$^{fl/-}$ *Atm*$^{-/-}$ males, because in the latter model chromosome axes were highly fragmented at early pachynema.

**Atr deletion does not influence DSB abundance**. We further examined roles of ATR in DSB formation. Orthologs of ATM influence DSB homeostasis by acting as negative regulators of DSB induction[38–42]. In *Saccharomyces cerevisiae* ATR promotes DSB formation indirectly by increasing the length of prophase I[43], but its impact on DSB levels in mammals is unknown. To address this question, we measured abundance of covalent SPO11-oligonucleotide (SPO11-oligo) complexes, by-products of DSB induction, in testis extracts[44]. Consistent with previous work[40], in *Atm*$^{-/-}$ testes SPO11-oligo complex levels were elevated relative to those observed in wild-type mice (Fig. 4a, b). However, SPO11-oligo complex levels in *Atr*$^{fl/-}$ testes, as well as in *Atr*$^{fl/+}$ males without the *Ngn3-Cre* transgene, were not detectably changed relative to controls (Fig. 4a, b). SPO11-oligo complex levels in *Atr*$^{fl/-}$ *Atm*$^{-/-}$ testes were similar to those in *Atm*$^{-/-}$ testes ($p = 0.29$; Fig. 4a, b). Thus, ATR and ATM have distinct functions with respect to DSB formation.

In addition to its role in DSB homeostasis, ATM is implicated in nucleolytic processing of DSBs[40, 45–47]. SPO11 oligos are longer in *Atm*$^{-/-}$ mice, with an increase in intermediate (~40–70 nucleotide (nt)) and large (>300 nt) species at the expense of the small (~15–27 nt) ones observed in wild type[40]. SPO11-oligo size distribution in *Atr*$^{fl/-}$ males and in *Atr*$^{fl/+}$ males without the *Ngn3-Cre* transgene were not detectably changed relative to controls (Fig. 4c, d). However, in *Atr*$^{fl/-}$ *Atm*$^{-/-}$ testes, SPO11 oligos were on average even larger than in *Atm*$^{-/-}$ testes, with a decrease in the abundance of small and intermediate oligos and a substantial increase in the amount of high-molecular-weight species running near the top of the gel (Fig. 4c, d). We conclude that ATR is largely dispensable for initial nucleolytic processing of SPO11-generated DSBs, but that it can partially compensate when ATM is not present to promote these events.

**Atr controls RAD51 and DMC1 abundance at DSBs in leptonema**. During leptonema, resected DSBs are coated with RPA (replication protein A) and the recombinases RAD51 and DMC1, which carry out strand invasion and recombinational repair. We used immunostaining to determine whether these early recombination components are influenced by ATR. Leptotene focus counts for RPA subunit 2 (hereafter termed RPA) were similar between *Atr*$^{fl/-}$ and controls (Fig. 4e, f). This finding supports conclusions from SPO11-oligo complex quantification that ATR does not influence DSB abundance (Fig. 4a, b), and is consistent with RPA acting upstream of ATR[48]. In addition, RPA counts in controls did not differ from those in *Atr*$^{fl/+}$ males without the *Ngn3-Cre* transgene (Fig. 4 legend). Importantly though, in *Atr*$^{fl/-}$ males, RAD51 counts were reduced by almost half relative to controls (Fig. 4g, h). DMC1 counts were lower by a similar magnitude (Fig. 4i, j). Thus, in *Atr*$^{fl/-}$ males, DSB abundance appears grossly unaffected, but recombinase localization is compromised.

We also quantified RPA2, RAD51 and DMC1 foci at leptonema in *Atm*$^{-/-}$ and *Atr*$^{fl/-}$ *Atm*$^{-/-}$ males. In both models, RPA counts were elevated twofold relative to *Atr*$^{fl/-}$ and control males (Fig. 4e, f). In *Atm*$^{-/-}$ males, RAD51 and DMC1 focus counts were similar to controls (Fig. 4g–j). The failure of

**Fig. 3** ATR is required for homologous synapsis. Examples of early pachytene synaptic outcomes in **a** *Atr*$^{fl/+}$ males ($n = 3$ males) and **b** *Atr*$^{fl/-}$ males ($n = 3$ males) assessed using HORMAD2 (magenta), SYCP3 (green) and subsequent DNA FISH using *Slx* and *Sly* probes (labeled in insets as X chromosome in green and Y chromosome in white). The *Slx* probe hybridizes to a sub-region of the X chromosome (circled), while the *Sly* probe coats the majority of the Y chromosome. In *Atr*$^{fl/+}$ males (**a**), both the autosomes and the XY PARs are synapsed, while the non-homologous regions of the XY pair are asynapsed. In *Atr*$^{fl/+}$ males ($n = 104$ cells, 2 males), 90 cells had normal synapsis, 12 cells had asynapsis of both the XY and autosomes, 1 cell had asynapsis only of the XY and 1 cell had asynapsis only of the autosomes. **b** Three examples of synaptic defects in *Atr*$^{fl/-}$ males, each described above respective image. In *Atr*$^{fl/-}$ males ($n = 107$ cells, 2 males), 21 cells had normal synapsis, 59 cells had asynapsis of both the XY and autosomes, 23 cells had asynapsis only of the XY, 4 cells had asynapsis only of the autosomes, 10 cells had X self-synapsis, 11 cells had Y self-synapsis and 7 had non-homologous synapsis between the X and/or Y and autosomes. **c, d** Comparison of pachytene synaptic outcomes in **c** *Spo11*$^{-/-}$ and **d** *Atr*$^{fl/-}$ *Spo11*$^{-/-}$ males using immunostaining for SYCE2 (magenta) and SYCP3 (green). **e** Quantitation of complete, partial and focal SC in *Spo11*$^{-/-}$ males ($n = 2$ males; 88 cells) and *Atr*$^{fl/-}$ *Spo11*$^{-/-}$ males ($n = 2$ males; 96 cells). Means and $p$ values (unpaired $t$-test) indicated. **f–h** Epistasis analysis of SPO11 and ATR in synapsis using the same markers plus immunostaining for centromeres (CENT; white) to determine CS number. For each cell, colocalizing SYCE2-CENT signals are indicated with dashed circles, and resulting CS numbers are shown. **i** CS number in *Atr*$^{fl/+}$ males ($n = 2$ males; 55 cells), *Atr*$^{fl/-}$ males ($n = 2$ males; 37 cells), *Spo11*$^{-/-}$ males ($n = 2$ males; 114 cells) and *Atr*$^{fl/-}$ *Spo11*$^{-/-}$ males ($n = 2$ males; 64 cells). Mean values and $p$ values (Mann–Whitney test) indicated. Scale bars 10 μm

recombination focus counts to fully account for the large increase in DSB formation in the absence of ATM has been proposed to reflect inability of cytological methods to resolve the clustered DSBs that form nearby on the same chromatid or pair of sister chromatids[40, 42]. However, in $Atr^{fl/-} Atm^{-/-}$ males RAD51 and

DMC1 counts were reduced to levels even lower than that observed in $Atr^{fl/-}$ males (Fig. 4g–j). Thus, RPA counts are increased by deleting ATM but not ATR, while RAD51 and DMC1 counts are decreased by deleting ATR, and even more so by deleting both ATR and ATM.

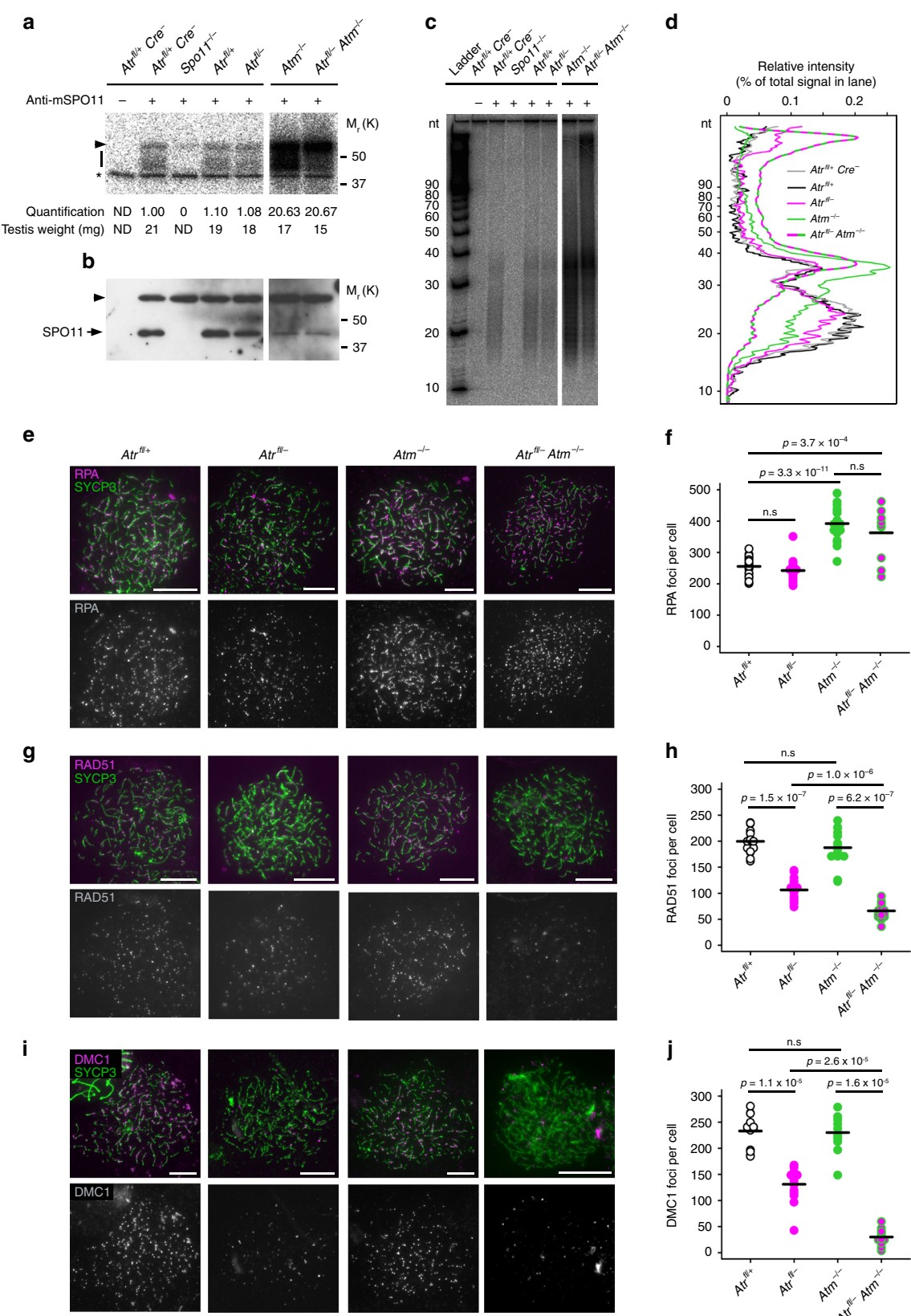

***Atr* regulates DSB dynamics during later prophase I**. We also tested if ATR influences later stages of recombination by quantifying RPA, RAD51 and DMC1 foci throughout prophase I. We restricted our analysis to *Atr*<sup>fl/−</sup> males because the extensive chromosome fragmentation after leptonema in *Atr*<sup>fl/−</sup> *Atm*<sup>−/−</sup> testes prevented a comparison of ATR and ATM functions at these later stages. Interestingly, while RPA counts in *Atr*<sup>fl/−</sup> males were equivalent to those in controls at late leptonema (Fig. 4e, f), they were reduced at mid zygonema (Fig. 5a–c). RAD51 and DMC1 counts were also lower in *Atr*<sup>fl/−</sup> than control males at this stage (Fig. 5c; Supplementary Fig. 3a-d). At early pachynema, DSB markers were assayed both on the autosomes and on the X chromosome, focusing initially on cells with normal autosomal synapsis. Relative to control males, in *Atr*<sup>fl/−</sup> males RPA, RAD51 and DMC1 counts were reduced on the autosomes (Fig. 5d–f; Supplementary Fig. 3e-h) and on the X chromosome (Fig. 5g, h, j; Supplementary Fig. 3i-l). The recombination defect in *Atr*<sup>fl/−</sup> males therefore affects RAD51 and DMC1 at leptonema, and all three DSB markers at mid zygonema and early pachynema.

In mice, synapsis is dependent upon recombination[35, 36, 49, 50]. Asynapsis in *Atr*<sup>fl/−</sup> males could therefore result not only from SC defects per se (Fig. 3), but also from aberrant recombination. We therefore asked whether early pachytene *Atr*<sup>fl/−</sup> spermatocytes with autosomal asynapsis exhibit alterations in DSB marker counts relative to *Atr*<sup>fl/−</sup> spermatocytes with normal autosomal synapsis. We used RPA as a representative DSB marker, and determined counts on the X chromosome as an indication of recombination levels. Since it can be obscured by asynapsed autosomes, the X chromosome was identified using *Slx* DNA FISH. In *Atr*<sup>fl/−</sup> cells with autosomal asynapsis (Fig. 5i) X-chromosome RPA counts were lower than those in *Atr*<sup>fl/−</sup> cells exhibiting normal autosomal synapsis (Fig. 5h, j). Greater alterations in recombination markers therefore correlate with the asynapsis phenotype in *Atr*<sup>fl/−</sup> males.

To further define ATR roles later in recombination, we examined the intermediate recombination marker RNF212, a RING-family E3 ligase that localizes to sites of synapsis and is implicated in designation of crossovers[51]. Interestingly, RNF212 focus counts were higher in *Atr*<sup>fl/−</sup> cells relative to controls (Fig. 6a–d). The elevation was observed in cells with normal synapsis as well as those with autosomal asynapsis. Thus, while deletion of ATR causes a reduction in RPA, RAD51 and DMC1 counts at early pachynema, it leads to an elevation in RNF212 counts at this stage.

## Discussion

The functions of ATR in mammalian meiosis have been unclear. We show here that ATR is required during unperturbed meiosis to regulate chromosome axis integrity, synapsis and recombination. *Atr* haploinsufficiency compromises the DNA damage response during mitosis[13]. We did not observe haploinsufficiency phenotypes in male meiosis, possibly because they are too mild to detect by our approaches. Alternatively, since ATR expression in the testis far exceeds that in other tissues[21], *Atr* haploinsufficiency may be better tolerated during meiosis than mitosis. As is the case in mitosis[1], in meiosis ATR has roles that are both shared and distinct from ATM. Differences in ATR and ATM functions are likely explained by the contrasting substrate specificities[1] and meiotic expression profiles[24] of these kinases.

Deletion of *Atr* causes mid-pachytene germ cell elimination. This phenotype is also observed in mice deficient in all three DNA damage-regulated PIKKs. Multiple lines of evidence suggest that PIKK depletion in these mice is efficient. Nevertheless, the use of conditional alleles means that residual PIKK activity may be present and sufficient for checkpoint maintenance. Setting aside this caveat, our findings do not exclude a contribution of PIKKs to mid-pachytene elimination in synapsis and recombination mutants. However, they do confirm the existence of additional mechanisms that can trigger elimination. Under such circumstances, mid-pachytene failure is likely caused by defective MSCI[27]. We suggest that the coexistence of multiple overlapping surveillance mechanisms during prophase I in males explains why checkpoint responses are more robust than those in females[52, 53].

Like ATM[54–56], ATR regulates homologous synapsis. ATR can promote synapsis independently of meiotic DSB formation, presumably through modification of SC proteins. Among the many SC components, HORMAD1/2 and SMC3 are established ATR phosphortargets[16, 57, 58]. While additional SC candidates no doubt exist, HORMAD1 is of particular interest, because this protein can also promote synapsis in the absence of recombination[34]. We find that in wild-type males, asynapsis of the XY pair occurs at similar frequency to that of all autosomal pairs combined. This bias towards XY asynapsis may be attributable to the small length and terminal location of the PAR. Notably however, in *Atr* mutants, the bias is exaggerated, such that asynapsis of the XY bivalent occurs at higher frequency than that of all autosomes combined. The PAR is unusual, undergoing late DSB formation and synapsis[33], and being enriched for repressive chromatin marks not observed at the termini of autosomal bivalents[59, 60]. We suggest that ATR promotes one or more of these unique properties to ensure successful XY interactions.

DSB frequency increases drastically when ATM is missing[40], but ATR deficiency has little if any effect on SPO11-oligo complex amounts in either ATM-proficient or ATM-deficient backgrounds. Thus, we conclude that ATR does not contribute significantly to control of DSB numbers. *Atm* mutants also exhibit an increase in the lengths of SPO11 oligos[40].

**Fig. 4** ATR ablation does not alter DSB levels but leads to reduction in leptotene recombinase counts. **a–d** Analysis of SPO11-oligo complexes in P13 testes. SPO11 was immunoprecipitated from whole-testis extracts and SPO11-associated oligos were end-labeled with terminal deoxynucleotidyl transferase and [α-<sup>32</sup>P] dCTP, then either separated on SDS-PAGE gels followed by autoradiography (**a**) and western blotting with anti-SPO11 antibody (**b**) or digested with proteinase K and resolved on denaturing polyacrylamide sequencing gels (**c**; background-subtracted lane traces in **d**). A representative experiment is shown. An additional *Atr*<sup>fl/+</sup> *Cre−* control is shown to demonstrate that the *Ngn3-Cre* transgene does not influence SPO11-oligo levels. In **a**, **b**, the bar indicates SPO11-oligo complexes, arrowhead indicates the immunoglobulin heavy chain and asterisk marks non-specific labeling; ND not determined. Each panel shows lanes from the same exposure of a single western blot or autoradiograph, with intervening lanes omitted. For quantitation, SPO11-oligo complex signals were background-subtracted and normalized to *Atr*<sup>fl/+</sup> *Cre-* (n = 2) controls. SPO11-oligo quantitation: *Atr*<sup>fl/+</sup> (1.05 ± 0.07-fold, mean and s.d., n = 2 males), *Atr*<sup>fl/−</sup> (1.17 ± 0.22-fold, n = 4 males), *Atm*<sup>−/−</sup> (14.58 ± 5.24-fold, n = 3 males) and *Atr*<sup>fl/−</sup> *Atm*<sup>−/−</sup> males (11.38 ± 13.15-fold). The reduced SPO11 protein levels in *Atm*<sup>−/−</sup> were previously documented[40, 79], but the molecular basis is not understood. **e–j** Analysis of leptotene focus counts using SYCP3 (green) and early recombination markers (magenta): RPA (**e**, **f**), RAD51 (**g**, **h**) and DMC1 (**i**, **j**) in *Atr*<sup>fl/+</sup> males (n = 2 males; 27 cells for RPA, 19 cells for RAD51, 14 cells for DMC1), *Atr*<sup>fl/−</sup> males (n = 2 males; 21 cells for RPA, 19 cells for RAD51, 13 cells for DMC1), *Atm*<sup>−/−</sup> males (n = 2 males; 22 cells for RPA, 15 cells for RAD51, 13 cells for DMC1) and *Atr*<sup>fl/−</sup> *Atm*<sup>−/−</sup> males (n = 2 males; 10 cells for RPA, 20 cells for RAD51, 13 cells for DMC1). RPA counts are not significantly different between *Atr*<sup>fl/−</sup> males (mean 249 foci) and *Atr*<sup>fl/+</sup> males without the *Ngn3-Cre* transgene (mean 227 foci, n = 2 males, 17 cells), and are thus not influenced by *Atr* haploinsufficiency. Mean and p values (Mann–Whitney test) indicated. Scale bars 10 μm

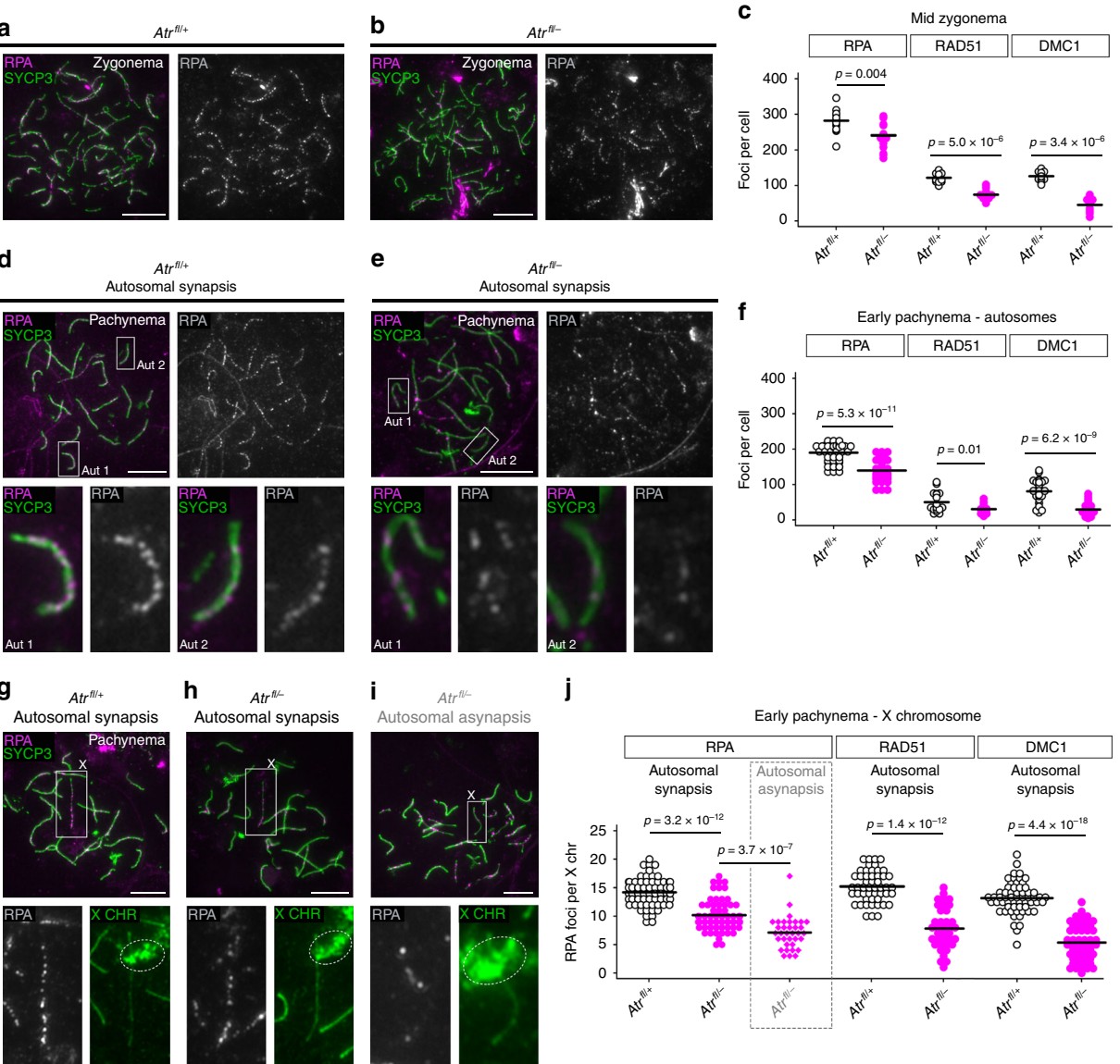

**Fig. 5** ATR regulates DSB marker counts during zygonema and pachynema. **a**, **b** RPA (magenta) and SYCP3 (green) immunostaining in $Atr^{fl/+}$ and $Atr^{fl/-}$ males. **c** RPA, RAD51 and DMC1 counts at mid zygonema in $Atr^{fl/+}$ males ($n = 2$ males; 16 cells for RPA, 15 cells for RAD51, 15 cells for DMC1) and $Atr^{fl/-}$ males ($n = 2$ males; 16 cells for RPA, 15 cells for RAD51, 15 cells for DMC1). **d**, **e** Examples of early pachytene cells from $Atr^{fl/+}$ and $Atr^{fl/-}$ males with normal autosomal synapsis. Two representative autosomes (aut 1 and 2) are boxed in upper panels and magnified in lower panels. **f** autosomal RPA, RAD51 and DMC1 counts at early pachynema in $Atr^{fl/+}$ males ($n = 2$ males; 49 cells for RPA, 19 cells for RAD51, 33 cells for DMC1) and $Atr^{fl/-}$ males ($n = 2$ males; 43 cells for RPA, 19 cells for RAD51, 35 cells for DMC1). **g**, **h** Early pachytene $Atr^{fl/+}$ and $Atr^{fl/-}$ cells with normal autosomal synapsis and the X chromosome (boxed in upper panels) identified by *Slx* DNA FISH (dashed circles in lower panels). **i** Early pachytene $Atr^{fl/-}$ cell with autosomal asynapsis. **j** X-chromosome RPA, RAD51 and DMC1 counts at early pachynema in $Atr^{fl/+}$ males ($n = 2$ males; 64 cells for RPA, 49 cells for RAD51, 59 cells for DMC1) and $Atr^{fl/-}$ males ($n = 2$ males; 62 cells for RPA in autosomal synapsis category, 35 cells for RPA in autosomal asynapsis category, 36 cells for RAD51, 53 cells for DMC1). Mean and $p$ values (Mann–Whitney test) indicated. Scale bars 10 μm

This phenotype is not observed in *Atr* single mutants, but SPO11-oligos become even longer in *Atr Atm* double mutants. These findings suggest that ATR can partially substitute for ATM in promoting normal nucleolytic processing of DSBs. For example, ATR may influence the 3′→5′ exonuclease activity of MRE11, which is known to govern Spo11-oligo length in yeast[61]. CtIP and its yeast ortholog Sae2, MRE11 partners important for DSB processing, are PIKK phosphortargets[62–64].

Although ATR deficiency does not appear to modulate DSB numbers, *Atr* mutants do display a substantial reduction in RAD51 and DMC1 focus counts. This finding suggests that ATR specifically promotes the assembly of RAD51 and DMC1 on resected DSBs. It has been suggested that ATR phosphorylates RAD51[65]. ATR also phosphorylates CHK1 during meiosis[66], and CHK1 in turn promotes RAD51 loading at DSBs[67]. A non-exclusive alternative is that focus numbers are reduced in the mutant because ATR slows their turnover, for example by inhibiting use of the sister chromatid as a recombination partner[5, 68]. *Atm* single mutants do not display a reduction in RAD51 and DMC1 foci, but the interpretation of this result is complicated by the fact that the mutants make more DSBs that are thought to often occur in clusters that may not be cytologically distinguishable from single DSBs[40, 42]. Thus, whether and how ATM might influence RAD51 and DMC1 assembly has been unclear.

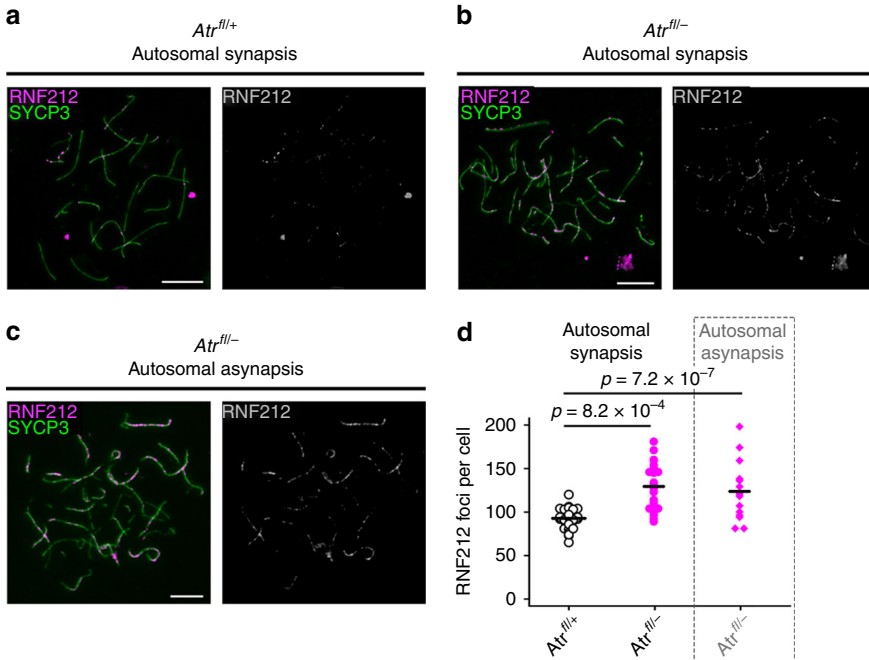

**Fig. 6** ATR regulates RNF212 counts during pachynema. **a–c** RNF212 (magenta) and SYCP3 (green) immunostaining at early pachynema in $Atr^{fl/+}$ and $Atr^{fl/-}$ males. **d** RNF212 counts in $Atr^{fl/+}$ male ($n = 1$ male; 29 cells) and $Atr^{fl/-}$ male with normal autosomal synapsis ($n = 1$ male; 15 cells) or autosomal asynapsis ($n = 1$ males; 25 cells). Mean and $p$ values (Mann–Whitney test) indicated. Scale bars 10 μm

We find that RAD51 and DMC1 focus counts are lower in the *Atr Atm* double mutants than in the *Atr* single mutant without an apparent change in DSB number. This finding suggests that ATM can indeed facilitate assembly of RAD51- and DMC1-containing recombination intermediates, at least in the absence of ATR. If so, ATM may promote RAD51 and DMC1 assembly directly, or may do so indirectly by fostering DSB resection, as its ortholog Tel1 does in yeast[2, 46]. ATM may also influence the lifespan of RAD51 and DMC1 foci via effects on sister chromatid recombination.

Later in prophase I, spermatocytes lacking ATR exhibit lower counts not only for RAD51 and DMC1 but also for RPA. In contrast, focus counts for the intermediate recombination marker RNF212 are elevated. These changes could imply additional functions for ATR in later processing of recombination intermediates and/or crossover designation. Alternatively, the reduction in RPA, RAD51 and DMC1 may indicate the presence of fewer unrepaired DSBs. This possibility is not precluded by the RNF212 findings, because RNF212 can localize to the SC in the absence of recombination[51]. Reduced DSB number in *Atr* mutants could result from premature repair using the sister chromatid[5], or from a failure to induce new DSBs on asynapsed chromosomes[69]. Interestingly, the ATR substrates HORMAD1/2 are implicated both in inhibiting sister chromatid recombination[70, 71] and in promoting ongoing DSB formation[70, 72]. Our data are less consistent with a defect in generating new DSBs on asynapsed chromosomes, because SPO11-oligo data showed no reduction in DSB formation in *Atr* mutants. Furthermore, in these mutants decreased RPA counts were also observed on synapsed autosomes, and the magnitude of the decrease was similar to that seen for the asynapsed X chromosome. If there are fewer DSBs during pachynema in the *Atr* mutant, then the increase in RNF212 foci seen at this stage could imply a role for ATR in regulating RNF212 abundance at the SC independently of recombination.

Whether ATR regulates crossover formation and designation is currently unclear. Analyses of male mice with a hypomorphic *Atr* mutation or with pharmacologically inhibited ATR suggest that

this kinase is required for crossovers[73]. We could not analyze crossover formation in our *Atr* mutants because crossover markers appear after the point of germ cell elimination in these mice (Supplementary Fig. 4). Conditional deletion of *Atr* in the female mouse germline, where recombination defects incur germ cell elimination later in meiosis[74], may help to resolve this point.

## Methods

**Animal experiments**. All mice were maintained under UK Home Office Regulations, UK Animals (Scientific Procedures) Act 1986, and according to ethical guidelines at the National Institute for Medical Research (NIMR) and the Francis Crick Institute Mill Hill laboratory. Permission for animal experiments was granted by The Crick Biological Research Facility Strategic Oversight Committee (BRF-SOC) incorporating Animal Welfare and Ethical Review Body (AWERB) (Project Licence P8ECF28D9 granted by the Secretary of State). Genetically modified models are previously published and are maintained on a predominantly C57BL/6 background: *Ngn3-Cre*[20], *Stra8-Cre*[18], *Atr*[14], *Atr flox*[75], *Atm*[76], *Atm flox*[31], *Spo11*[36] and *Dmc1*[49]. *Prkdc*[scid/scid] mice were obtained from the Jackson Labs and are maintained on an NOD background. Littermate controls were used where possible.

**Immunofluorescence, focus counting and western blotting**. Immunofluorescence experiments on surface spread spermatocytes were carried out as previously described[28]. In brief, cells were permeabilized for 10 min in 0.05% Triton X-100 and fixed for 1 h minimum in 2% formaldehyde, 0.02% SDS in phosphate-buffered saline (PBS). Slides were rinsed in distilled water, air dried and blocked in PBT (0.15% bovine serum albumin, 0.10% TWEEN-20 in PBS) for 1 h. Slides were incubated with the following antibodies in a humid chamber overnight at 37 °C: guinea-pig anti-SYCP3 (made in-house) 1:500, rabbit anti-SYCP3 (Abcam ab-15092) 1:100, rabbit anti-ATR (Cell Signalling #2790) 1:50, mouse anti-γH2AX (Millipore 05–636) 1:100, rabbit anti-HORMAD2 (Tóth lab) 1:100, guinea-pig anti-SYCE2 (gift from Howard Cooke/Ian Adams) 1:800, human anti-centromere (CREST ab gift from Bill Earnshaw) 1:1000, rabbit anti-RPA (Abcam ab-2175) 1:100 anti-rabbit RAD51 (Calbiochem PC130) 1:25, goat anti-DMC1 (Santa Cruz sc-8973) 1:25. Western blotting was carried out as previously described[77] (Supplementary Fig. 6). Counting of recombination protein foci was performed manually, considering only foci that colocalized with axial elements.

**Meiotic staging**. The presence of asynapsis can make discrimination between zygotene cells and pachytene cells with asynapsis challenging. We used the following criteria (Supplementary Fig. 5). (1) DAPI (4′,6-diamidino-2-phenylindole) staining: at zygonema DAPI staining is bright and centromeres are clustered in a

few subdomains, while in pachynema DAPI staining is heterogeneous, with euchromatin stained faintly and centromeres stained brightly and forming multiple subdomains. (2) The length and thickness of axial elements/SCs: axial elements are long and thin in zygonema, and are not yet fully formed, but they are shorter and thicker and fully formed in pachynema. (3) Chromosomal asynchrony: in contrast to zygotene cells, pachytene cells with asynapsis exhibit asynchrony in synapsis between individual bivalents, i.e., the coexistence of completely asynapsed bivalents and multiple fully synapsed bivalents. For comparison of synapsis between *Spo11*$^{-/-}$ males and *Atr*$^{fl/-}$ *Spo11*$^{-/-}$ males, pachytene cells were identified by virtue of having fully developed axial elements.

**RNA, DNA FISH and SPO11-oligo analysis**. FISH was carried out with digoxigenin-labeled probes as previously described[78]. CHORI BAC probe, RP24-204O18 (CHORI) was used for *Scml2* RNA FISH, RP23-470D15 for *Slx* DNA FISH, and RP24-502P5 for *Sly* DNA FISH. Analyses of abundance of SPO11-oligo complexes and sizes of SPO11 oligos were performed as previously described[40, 44].

**Microscopy**. Imaging was performed using an Olympus IX70 inverted microscope with a 100-W mercury arc lamp. For chromosome spread and RNA FISH imaging, an Olympus UPlanApo 100×/1.35 NA oil immersion objective was used. For testis section imaging, an Olympus UPlanApo 40×/0.75 NA objective was used. A Deltavision RT computer-assisted Photometrics CoolsnapHQ CCD camera with an ICX285 Progressive scan CCD image sensor was utilized for image capture. Then, 8 or 16-bit (512 × 512 or 1024 × 1024 pixels) raw images of each channel were captured and later processed using Fiji software.

**Statistics**. Statistical calculations were performed using GraphPad Prism 6.0. For comparison of two genotypes, Mann–Whitney test or *t*-tests were performed.

**Data availability**. The data that support the findings of this study are available from the corresponding author upon request.

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

## Acknowledgements

We are grateful to Bill Earnshaw for providing the CREST anti-centromere antibody, Howard Cooke, Ian Adams for providing the SYCE2 antibody, Paula Cohen for providing the anti-MLH3 antibody, Neil Hunter for the anti-RNF212 antibody, Eric Brown for providing the *Atr* flox and null mice, members of the Turner laboratory for critical reading of the manuscript and the Francis Crick Institute Technology platforms for excellent assistance. This work was supported by the Francis Crick Institute which receives its core funding from Cancer Research UK (FC001193), the UK Medical Research Council (FC001193) and the Wellcome Trust (FC001193). Work in the Keeney lab is supported in part by the US National Institutes of Health grant R35 GM118092; J.L. was supported in part by American Cancer Society fellowship PF-12-157-01-DMC. S.P., A.M.-L. and I.R. are supported by the Ministerio de Ciencia e Innovación (BFU2016-80370-P).

## Author contributions

A.W., O.O., V.M. and J.M.A.T. performed animal generation and genotyping; A.W., S.K.M., M.S., S.P., A.M.-L. and J.M.A.T. performed immunofluorescence; A.W., E.E. and V.M. performed western blots; A.W. and S.K.M. performed RNA FISH; T.H. performed DNA FISH; J.L. and S.K. performed SPO11-oligo experiments; A.W. and D.d.R. performed histology; A.W. and J.Z. performed data plotting and statistics; A.T. supplied HORMAD2 antibody. J.M.A.T. wrote the manuscript with critical input from A.T., S.K., I.R. and Turner lab members.

## Additional information

**Competing interests:** The authors declare no competing interests.

