## [Peer Review File · Nature Communications]

Reviewers' comments:

Reviewer #1 (Remarks to the Author):

ATR is a multifunctional regulator of male mouse meiosis

Strengths/Highlights:

1. Novel role for ATR in meiotic recombination
2. Reinforces the importance of unpaired chromatin in the surveillance of meiosis.

Comments/Critique:

1. The authors conclude that ATR is required for the assembly of RAD51 and DMC1 at RPA coated sites (DSB resected sites). They also show that ATR functions in synaptonemal complex (SC) extension/elongation. I wonder whether changes in numbers of recombination foci seen upon the loss of ATR activity might be a direct consequence of defects in SC formation. It is not clear whether the reduction in foci translates to altered association with chromatin in general or to the mapped DSB sites.

This can be done by:

- a. By determining the levels of DMC1 and RAD51 in the soluble vs insoluble chromatin by a simple nuclear fractionation of control and mutant animals.
- b. RAD51 and DMC1 CHIP-PCR at a handful of mapped sites (Smagulova et al. 2011) in the control and mutant animals.

4. It would be interesting to know the dynamics of pRPA (S33) levels, a known signal for MSCI/MSUC that is ATR dependent (Fedoriw et al. 2015).

Similarly other markers of unpaired Chromatin such as pCHK1 (S317 & S345) can also be monitored in the mutants.

5. Does the loss of ATR result in altered levels of sex body associated repressive chromatin marks such as H3K9me2 and H2AK119ub1?

6. As HORMAD1 plays a role in SC formation and is phosphorylated (pS375) presumably by ATR (Fukuda et al. 2012), it might be a good idea to look at pHORMAD1 distribution along autosomal asynapsed axes upon the inhibition of ATR activity.

References

Fedoriw AM, Menon D, Kim Y, Mu W, Magnuson T. 2015. Key mediators of somatic ATR signaling localize to unpaired chromosomes in spermatocytes. *Development*.

<http://dev.biologists.org/cgi/doi/10.1242/dev.126078>
<http://www.ncbi.nlm.nih.gov/pubmed/26209650>.

Fukuda T, Pratto F, Schimenti JC, Turner JM a, Camerini-Otero RD, Höög C. 2012.

Phosphorylation of chromosome core components may serve as axis marks for the status of chromosomal events during mammalian meiosis. *PLoS Genet* 8.

Smagulova F, Gregoret I V, Brick K, Khil P, Camerini-Otero RD, Petukhova G V. 2011.

Genome-wide analysis reveals novel molecular features of mouse recombination hotspots.

Nature 472: 375–378.

Reviewer #2 (Remarks to the Author):

“ATR is a multifunctional regulator of male mouse meiosis”
by Widger et al.

In this manuscript, Widger et al. generate and analyze a meiosis-conditional KO allele of ATR, a major mammalian PIKK central to the DNA damage response. They further use mouse genetics to more finely dissect ATR-dependency of the observed spermatogenesis phenotypes. They establish ATR as an essential component for normal male meiosis, and shed some light on possible mechanisms for its requirement. This is a timely paper that complements other recent (yeast) studies that elucidate ATR function in meiosis. It is well-written and –structured.

Below are my specific comments and suggestions.

1. Use of *Atr* fl/+ as the only control mice. There are clearly some – albeit relatively mild – meiotic defects in these animals (e.g. autosomal asynapsis at early pachytene). For those analyses where this is clearly the case, the authors need to show fully wt controls (= age-matched *Atr*+/+ of the same strain background) for comparison.
2. Discuss evidence (or lack thereof) of ATR haploinsufficiency in meiosis, esp. in mice (but also relevant human studies). How do the control mice used in this study compare to published data on *Atr*+/- mice?
3. Discuss O’Driscoll et al. (AJHG, 2007) – overexpression of RPA under conditions of ATR haploinsufficiency.
4. Quantification of RPA foci (Fig. 3e-f).
 - a) Based on the two examples shown in the images, it seems possible that RPA foci are reduced in fl/- compared to fl/+. The fl/+ nucleus appears to have both bright and faint foci (and more total foci), whereas fl/- only has bright foci (and fewer). If this impression is accurate (= if the images chosen for Fig. 3e-f are representative), how were these differences in faint-bright appearance taken into account when foci were quantified? There is no mention in Methods about how RPA foci were counted – manually or not, was background adjusted for (which would affect counting of fainter but possibly genuine foci), co-localization requirement with axis?
 - b) Relating to my point 1 above, the authors need to perform the same analysis in fully wt controls.
Brighter foci in fl/- would be consistent with defects in exchanging RPA for RAD51/DMC1.
5. Intermediate markers of recombination need to be analyzed – MSH4/5 and/or RNF212.

Are these also reduced or does crossover homeostasis kick in at this point in fl/-?

6. CS score.

a) I cannot quite follow what it is that the CS score (Fig. 2f-i) adds to the analysis. How is this a "higher throughput quantitation" (lines 145-46)?

b) Lines 151-152: Need to distinguish between non-homologous and homologous synapsis here. Spo11 KO cells rarely achieve extensive synapsis anyhow. Consider this in light of Bisig et al.'s (PLoS Genet 2012) findings on centromere pairing.

7. Suppl. Fig2, histology in Spo11 Atr double KO – looks like there is accumulation of cells in this stage IV tubule. Comment? Interpretation?

Minor comments:

8. I am confused about the H2AFX (as opposed to H2AX) nomenclature used in this manuscript. According to UniProt, the gene in mice is called H2afx, but the protein is H2AX - > the phosphorylated form of the histone should be called gamma-H2AX, not gamma-H2AFX.

9. Some mention of human phenotypes (esp. fertility) related to ATR defects would be nice in the introduction.

10. Is there a complete spermatogenic block in fl/- mice – any elongating spermatids visible? Need some statement regarding this.

11. Figure 2e, cartoon with focal SC. Only one dot-like SC should be drawn, as more than one per (discernible) axis is rarely seen.

12. Figure 3b. Less SPO11 protein in fl/- than Atm-/- . Comment?

13. Need to show examples in the Supplement of the staging described in lines 431-445.

Reviewer #3 (Remarks to the Author):

This manuscript analyzes the function of ATR kinase in mouse spermatocytes using an Ngn3-Cre construct, which allows a more efficient tissue-specific knockout than previous approaches. The authors confirm previous observations showing a role for ATR in sex chromosome inactivation and identify additional meiotic roles of ATR. They show that ATR mutants have defects in chromosome synapsis that are independent of meiotic DNA double-strand break (DSB) formation. Moreover, they show that focus formation of the DSB repair enzymes Rad51 and Dmc1 is reduced despite normal levels of DSB formation. In a genetic tour de force the authors furthermore inactivated the three DNA-damage-associated PIK kinases, ATM, ATR, and PRKDC. They find that inactivation of PRKDC removes the nucleus-wide DNA damage histone mark gammaH2AFX, which accumulates in Atr(fl/-), Atm(-/-) and Atr Atm double mutants. However, the triple mutant does not allow progression past the

typical stage IV arrest, indicating that the spermatocyte culling at this point relies on other mechanisms.

This is a clear and well-written manuscript with conclusions that are supported by the presented data. The finding that ATR mutants have normal DSB levels but reduced levels of repair foci is interesting and suggests possible similarities with the role of meiotic ATR in other organisms. My main issue with this study is the very limited analysis of the double and triple mutants. Overall this topic is of substantial interest to the field and with some additional data is expected to be of interest to a broad readership.

1. The authors went through the extremely time-consuming exercise of creating double and triple mutants of the PI3like kinases but then only analyze testis sections and the distribution of gammaH2AFX on spreads nuclei. Given the partially overlapping roles and substrates among these kinases, the authors should at least provide a commentary on the chromosomal phenotypes of these mutants. The single images provided in Figure 5 (c and e) do not allow adequate assessment of synaptic defects and are not discussed in this context. The authors should provide quantification of synaptic defects and of RAD51 or DMC1 foci in these mutants and relate them to the observations of the *Atr(fl/-)* phenotype presented in the first part of the manuscript.

2. P.5: Please clarify the incidence of self synapsis shown in figure 2B. The text specifically refers to 25% of *Atr* cells showing non-homologous synapsis between sex chromosomes and autosomes. If self synapsis was part of this count the classification should be rephrased accordingly.

Response to Reviewers: NCOMMS-17-12207A

We thank the Reviewers for their helpful comments. We would like to draw attention to two major additions:

- 1) As requested by Reviewer 2, we performed analyses in a second control genotype, in which males are wild type rather than heterozygous for *Atr*.
- 2) As requested by Reviewer 3, we performed additional meiotic analyses in *Atr Atm* double mutants. This change necessitated a reorganization of the manuscript: the section on stage IV arrest in *Atr Atm* double mutants has now been moved from the end to the beginning of the manuscript.

Please note that the responses below refer to the version of the manuscript deposited in Supplementary information, named ManuscriptTrackedChanges.docx. In that version we have tracked all changes for the convenience of the reviewers.

Reviewer 1:

1) The authors conclude that ATR is required for the assembly of RAD51 and DMC1 at RPA coated sites (DSB resected sites). They also show that ATR functions in synaptonemal complex (SC) extension/elongation. I wonder whether changes in numbers of recombination foci seen upon the loss of ATR activity might be a direct consequence of defects in SC formation...

Response: In mice, recombination initiates prior to, and is essential for, homologous synapsis (Mahadeveiah et al, *Nat Gen*, 2001, **27**:271; Romanienko et al, *Mol Cell* 2000, **6**:975; Baudat et al, *Mol Cell*, 2000, **6**:989). Moreover, SC defects are known to cause an increase in RAD51 focus numbers because SC formation is normally associated with down regulation of DSB formation (Kauppi et al, *Genes Dev*, 2013, **27**: 873; Wojtasz et al, *PLoS Genet*, 2009, **5**: e1000702). Finally, we document that ATR deficiency reduces focus numbers in leptotene cells, i.e., before any SC formation could have occurred. It is therefore not possible that the changes in numbers of recombination foci are a consequence (direct or indirect) of defects in SC formation.

.....it is not clear whether the reduction in foci translates to altered association with chromatin in general or to the mapped DSB sites. This can be done by a) By determining the levels of DMC1 and RAD51 in the soluble vs insoluble chromatin by a simple nuclear fractionation of control and mutant animals. b. RAD51 and DMC1 CHIP-PCR at a handful of mapped sites (Smagulova et al. 2011) in the control and mutant animals.

Response: It is not clear to us what scenario the reviewer has in mind with this suggestion. It is well established in the field that foci of RAD51 and DMC1 assemble specifically at sites of SPO11-dependent DSBs during meiosis. Prior studies have shown that focus formation is almost completely eliminated in the absence of SPO11-mediated DSBs (Romanienko et al, *Mol Cell* 2000, **6**:975; Baudat et al, *Mol Cell*, 2000, **6**:989) with the very small number of residual foci suggested to be the result of transposon-mediated DNA damage (Carofiglio et

al, *PLoS Genetics* 2013, **9**:e1003538). Thus, it is well accepted that focal accumulation of RAD51 and DMC1 is at DSBs, and we are not aware of any studies implicating general association with chromatin, i.e., not at DSBs, as a meaningful contributor to cytologically observable complexes of these proteins in mice. Our data clearly demonstrate that fewer foci are present early in prophase, whereas SPO11-oligo complexes are not reduced and RPA foci are elevated. These data are thus strong *prima facie* evidence that SPO11-dependent DSBs form relatively normally but assembly of DMC1 and RAD51 at these sites is impaired.

Importantly, we emphasize that all available data from many researchers clearly tie RAD51 and DMC1 foci to sites of DSBs within individual cells, not simply to “mapped DSB sites” in the genome. Mapped DSB sites (known as hotspots) are a population-average measure of the targeting of the DSB machinery by PRDM9. It is immaterial for our purposes whether the DSBs in the *Atr* mutant map to the same genomic coordinates as in wild type, since the defect we are examining pertains to what happens at individual DSBs once they have formed. Moreover, the *Atr* mutant does not resemble mutants lacking PRDM9, so there is no reason at present to consider altered DSB site choice as a candidate to explain our findings. The ChIP experiments proposed would be a substantial undertaking but would not, in our opinion, contribute to the understanding of the roles of ATR in promoting RAD51/DMC1 focus formation, so we have elected not to perform these experiments.

2) *It would be interesting to know the dynamics of pRPA (S33) levels, a known signal for MSCI/MSUC that is ATR dependent (Fedoriw et al. 2015). Similarly, other markers of unpaired Chromatin such as pCHK1 (S317 & S345) can also be monitored in the mutants.*

Response: We acquired and tested the antisera against pRPA (S33) and pCHK1 (S317 and S345) reported in the Fedoriw study, but we cannot reproduce the reported localization patterns. Furthermore, a recent manuscript from the Namekawa lab (PMID: 29360988) demonstrates that CHK1 does not localize to the XY pair in meiosis, and that the pCHK1 (S345) antibody used in the Fedoriw study is non-specific, giving equivalent staining patterns in wild type and *Chk1* knockout meiotic cells.

3) *Does the loss of ATR result in altered levels of sex body associated repressive chromatin marks such as H3K9me2 and H2AK119ub1?*

Response: H3K9me2 accumulates on the sex body from late pachynema onwards (Khalil et al, *PNAS*, 2004, **101**:16583; Namekawa et al, *Curr Biol*, 2006, **16**:660). Given that germ cells in our *Atr* mutant are eliminated earlier, at mid pachynema, this histone modification cannot be assayed. We have already published that loss of ATR prevents H2AK119ub1 accumulation on the sex body (Royo et al, *Genes Dev*, 2013, **27**:1484).

4) *As HORMAD1 plays a role in SC formation and is phosphorylated (pS375) presumably by ATR (Fukuda et al. 2012), it might be a good idea to look at pHORMAD1 distribution along autosomal asynapsed axes upon the inhibition of ATR activity.*

Response: We have published that phosphorylation of HORMAD1 at S375 on the asynapsed X and Y is dependent on ATR (Royo et al, *Genes Dev*, 2013, **27**:1484). Unfortunately, the

antibody used in that study, generated by the Höög lab and originally described in (Fukuda et al. *PLoS Genetics* 2012, 8:e1002485) no longer exists, so we cannot specifically assess localization to asynapsed autosomes in the *Atr* mutant.

Reviewer 2:

Major comments

1) *Use of Atr fl/+ as the only control mice. There are clearly some – albeit relatively mild – meiotic defects in these animals (e.g. autosomal asynapsis at early pachytene). For those analyses where this is clearly the case, the authors need to show fully wt controls (= age-matched Atr+/+ of the same strain background) for comparison.*

Response: Our control *Atr^{fl/+} Cre⁺* males do not exhibit meiotic defects. We demonstrate this by including a new analysis of *Atr^{fl/+}* males that do not carry the *Cre* transgene, and thus have wild type ATR levels. In these males, testis weights (page 3, lines 76-78), frequency of asynapsis (page 7, lines 160-162), SPO11-oligo complex levels (page 9, line 219 and 220), SPO11-oligo length (page 9, line 227-229) and RPA counts (page 10, line 243-245) are similar to *Atr^{fl/+} Cre⁺* males. To specifically address the reviewer's point about asynapsis; there is no statistically significant difference between the frequency of asynapsis in *Atr^{fl/+} Cre⁺* males (affecting 14% of early pachytene cells), and the *Atr^{fl/+} Cre* transgene negative males (affecting 14% of early pachytene cells). We nevertheless thank the reviewer for alerting our attention to published work that *Atr* heterozygosity causes mitotic phenotypes, so we have included this point on page 3, lines 57-60. Our findings that *Atr* heterozygosity does not cause meiotic phenotypes is discussed on page 12, lines 299-303.

2) *Discuss evidence (or lack thereof) of ATR haploinsufficiency in meiosis, esp. in mice (but also relevant human studies). How do the control mice used in this study compare to published data on Atr+/- mice?*

Response: As requested by the Reviewer, we now include a discussion of the contrasting effects of *Atr* heterozygosity on mouse meiosis (our findings) versus mitosis on page 12, lines 299-303.

3) *Discuss O'Driscoll et al. (AJHG, 2007) – overexpression of RPA under conditions of ATR haploinsufficiency.*

Response: We were unable to locate in this O'Driscoll manuscript information linking ATR haploinsufficiency to RPA overexpression. The manuscript is nevertheless relevant to our work and so we have cited it on page 3, lines 57 and 58, and page 12, lines 299 and 300.

4) *Quantification of RPA foci (Fig. 3e-f). a) Based on the two examples shown in the images, it seems possible that RPA foci are reduced in fl/- compared to fl/+. The fl/+ nucleus appears to have both bright and faint foci (and more total foci), whereas fl/- only has bright foci (and fewer). If this impression is accurate (= if the images chosen for Fig. 3e-f are representative), how were these differences in faint-bright appearance taken into account when foci were quantified? There is no mention in Methods about how RPA foci were counted – manually or not, was background adjusted for (which would affect counting of fainter but possibly genuine foci), co-localization requirement with axis? b) Relating to my point 1 above, the authors need to perform the same analysis in fully wt controls. Brighter foci in fl/- would be consistent with defects in exchanging RPA for RAD51/DMC1.*

Response: a) In the figure the *Atr fl/+* cell contains 261 RPA foci and the *Atr fl/-* cell 225 foci. RPA counts vary between cells, but the mean RPA counts, presented now in revised Fig. 4f, are not significantly different between these two genotypes. We count all foci that colocalize with SYCP3-positive axes, irrespective of whether the foci are bright or faint. A description of how we count foci is now added to the Methods, page 25, lines 593-594. b) as mentioned in our response to this reviewer's comment 1, RPA counts have now been performed on *Atr fl/+* males without the *Cre* transgene, page 10, lines 243-245.

5) *Intermediate markers of recombination need to be analyzed – MSH4/5 and/or RNF212. Are these also reduced or does crossover homeostasis kick in at this point in fl/-?*

Response: An analysis of RNF212 counts is now added (page 12, lines 289-295) and an accompanying new figure included (Figure 6). The findings are discussed on pages 15 and 16, lines 367-370 and 381-384.

6. *CS score. a) I cannot quite follow what it is that the CS score (Fig. 2f-i) adds to the analysis. How is this a “higher throughput quantitation” (lines 145-46)? b) Lines 151-152: Need to distinguish between non-homologous and homologous synapsis here. Spo11 KO cells rarely achieve extensive synapsis anyhow. Consider this in light of Bisig et al.'s (PLoS Genet 2012) findings on centromere pairing.*

Response: a) The CS score acts as an additional method to quantitate synapsis. The reviewer is correct that the term “higher throughput quantitation” is confusing, and so we have removed it. b) In response to reviewer's comment we have deleted the statement: “The mean CS number was also lower in *Spo11^{-/-}* than in *Atr^{fl/-}* males, demonstrating that synapsis is more severely impaired by loss of *SPO11* than loss of *ATR* (Fig. 2f,g,i). This statement is not relevant to the principal finding of the analysis, and it erroneously assumes that one can quantitatively compare non-homologous synapsis in the *Spo11* mutant to homologous synapsis in the *Atr* mutant.

7) *Suppl. Fig2, histology in Spo11 Atr double KO – looks like there is accumulation of cells in this stage IV tubule. Comment? Interpretation?*

Response: The apparent accumulation of cells is due to tangential sectioning of the tubule. This is explained in detail in Supp Figure legend 2, page 22, lines 528-533.

Minor comments:

8) *I am confused about the H2AFX (as opposed to H2AX) nomenclature used in this manuscript. According to UniProt, the gene in mice is called H2afx, but the protein is H2AX -> the phosphorylated form of the histone should be called gamma-H2AX, not gamma-H2AFX.*

Response: We have corrected the manuscript accordingly.

9) *Some mention of human phenotypes (esp. fertility) related to ATR defects would be nice in the introduction.*

Response: This is added on pages 2 and 3, lines 55-57.

10. *Is there a complete spermatogenic block in fl/- mice – any elongating spermatids visible? Need some statement regarding this.*

Response: There is a complete spermatogenic block with *Ngn3-Cre*, but some elongating spermatids were observed with the less robust deletion using *Stra8-Cre*. This information is now included on page 4, lines 85 and 86.

11. *Figure 2e, cartoon with focal SC. Only one dot-like SC should be drawn, as more than one per (discernible) axis is rarely seen.*

Response: Figure (which is now Figure 3) is amended accordingly.

12. Figure 3b. Less SPO11 protein in fl/- than *Atm*^{-/-}. Comment?

Response: We think the reviewer meant the opposite: SPO11 protein levels are lower in the *Atm* mutant (now Fig. 4b). This phenomenon was previously documented (Bellani et al, *Mol Cell Biol*, 2010, **30**:4391; Lange et al., *Nature*, 2011, **479**:237), but the molecular basis is not yet understood. We added citations to these references to the figure legend, page 20, lines 475-477.

13. Need to show examples in the Supplement of the staging described in lines 431-445.

Response: A new figure (Supp Figure 5) was added and referred to in the Methods section on page 25, line 600.

Reviewer 3:

1) *The authors went through the extremely time-consuming exercise of creating double and triple mutants of the PI3like kinases but then only analyze testis sections and the distribution of gammaH2AFX on spreads nuclei. Given the partially overlapping roles and substrates among these kinases, the authors should at least provide a commentary on the chromosomal phenotypes of these mutants. The single images provided in Figure 5 (c and e) do not allow adequate assessment of synaptic defects and are not discussed in this context. The authors should provide quantification of synaptic defects and of RAD51 or DMC1 foci in these mutants and relate them to the observations of the *Atr(fl/-)* phenotype presented in the first part of the manuscript.*

Response: We added analysis of the *Atr Atm* double KO, including more detailed histology (pages 5 and 6, lines 121-137), SPO11-oligo complex levels (page 9, lines 220-222; Fig. 4a,b), SPO11-oligo size (page 9 and 10, lines 224-235; Fig. 4c,d), and RPA, RAD51 and DMC1 counts at leptoneura (pages 10 and 11, lines 250-260). The new findings are discussed (pages 14 and 15, lines 335-364). Given that this mutant exhibits extensive axis fragmentation from zygonema on, analysis of synapsis and recombination markers later in prophase I is not possible. The triple *Atr Atm Prkdc* mutant is even more defective in axis fragmentation, which makes recombination foci counts of very limited value. Given that triple mutant samples are particularly difficult to acquire, we decided not to include extensive recombination foci counts from this model due to the very low benefit to cost ratio. This has been agreed by the editor.

2) *P.5: Please clarify the incidence of self synapsis shown in figure 2B. The text specifically refers to 25% of *Atr* cells showing non-homologous synapsis between sex chromosomes and autosomes. If self synapsis was part of this count the classification should be rephrased accordingly.*

Response: The statement has been reworded for clarity, page 7, lines 168-170 and quantitation of non-homologous sex chromosome – autosome synapsis and of self-synapsis of the X and of the Y is added to Figure Legend 3, page 19, lines 445-447.

REVIEWERS' COMMENTS:

Reviewer #1 (Remarks to the Author):

As stated in the original review, the strengths of this MS are a novel role for ATR in meiotic recombination and the reinforcement of the importance of unpaired chromatin in the surveillance of meiosis. Both points are of interest to the field.

The authors have presented mostly counterpoints to the first review with little additional new data. However, their responses are logical given the published data they quote and also the similarity in results with the second manuscript accompanying this one. Between the two manuscripts, the results are well substantiated and of importance to this field.

Reviewer #3 (Remarks to the Author):

The authors have addressed all the issues I raised in my previous review, although I ask that the increased axis fragmentation defect in the triple mutant (mentioned in the response to the reviewers) is also mentioned in the manuscript.

Minor things:

Line 103: "these cells" was a little confusing to me in the first read-through. I suggest spelling out that they are mid-pachytene Atr fl/- cells

Line 257: delete parenthesis

Response to Reviewers: NCOMMS-17-12207B (revised version)

We thank the Reviewers for their helpful comments. Please see below the responses.

Reviewer #1 (Remarks to the Author):

As stated in the original review, the strengths of this MS are a novel role for ATR in meiotic recombination and the reinforcement of the importance of unpaired chromatin in the surveillance of meiosis. Both points are of interest to the field.

The authors have presented mostly counterpoints to the first review with little additional new data. However, their responses are logical given the published data they quote and also the similarity in results with the second manuscript accompanying this one. Between the two manuscripts, the results are well substantiated and of importance to this field.

Response: As the reviewer did not have any requests, no response needed.

Reviewer #3 (Remarks to the Author):

1) The authors have addressed all the issues I raised in my previous review, although I ask that the increased axis fragmentation defect in the triple mutant (mentioned in the response to the reviewers) is also mentioned in the manuscript.

Response: As requested by reviewer, the increased axis fragmentation defect in the triple mutant is also mentioned in the manuscript (lines 175-177).

Reviewer #3 (Remarks to the Author):

Minor things:

2) Line 103: "these cells" was a little confusing to me in the first read-through. I suggest spelling out that they are mid-pachytene Atr fl/- cells

Response: As suggested by reviewer, the kind of cells were specified in line 123.

3) Line 257: delete parenthesis

Response: Parenthesis deleted and the change marked in line 286.